# Response of Extratropical Transitioning Tropical Cyclone Size to Ocean Warming: A Case Study for Typhoon Songda in 2016

Ziwei Miao and Xiaodong Tang *

Key Laboratory of Mesoscale Severe Weather, Ministry of Education, School of Atmospheric Sciences, Nanjing University, Nanjing 210023, China
* Correspondence: xdtang@nju.edu.cn

**Abstract:** This study attempts to investigate how future sea surface temperature increases will affect the size (radius of gale-force [17 m s$^{-1}$] wind at 10 m height; i.e., R17) evolution of tropical cyclones that undergo extratropical transition (ET) through sensitivity experiments of sea surface temperature (SST) for Typhoon Songda (2016) in the northwestern Pacific. Two numerical experiments were carried out, including a control simulation (control) and a sensitivity experiment (SST4.5) with SST increased by 4.5 degrees in the entire domain. The results showed that Songda tended to be stronger and larger with projected higher SSTs. Moreover, the momentum equation for tangential wind was utilized to study the mechanism of R17 evolution in different SST scenarios, in which the radial absolute vorticity flux term played a dominant role in generating a positive tendency of tangential wind. The results indicate that before ET, higher SSTs in the entire domain led to more active rainbands in both inner-core and outer-core regions. As a result, stronger secondary circulation and low-level inflow extended outward, and the absolute angular momentum (AAM) importing from the outer region increased, which led to a larger R17 in SST4.5. During the ET, the peripheral baroclinically driven frontal convection induced extensive boundary layer inflow, which accelerated the tangential flow in the outer frontal region through strong inward AAM transport. However, due to the lower latitude of the cyclone and the strong frontolysis at the outer side of the cold pool in SST4.5, the peripheral frontal convection reached the location of R17 later; thus, the increase in the cyclone size lagged behind that in the control.

**Keywords:** tropical cyclone; extratropical transition; size; sea surface temperature; momentum equation; baroclinic zone; frontal convection; cold pool



## 1. Introduction

A large number of tropical cyclones (TCs) undergo extratropical transition (ET), during which they move into the midlatitudes and interact with extratropical flow patterns [1–5]. An early climatological study indicated that the largest number of ET events occur in the western North Pacific, while the largest percentage (about 45%) of ET events occur in the North Atlantic basin [1]. Transitioning TCs often undergo a transformation from symmetric, warm-core to asymmetric, cold-core extratropical systems [1,6]. Although many TCs weaken under unfavorable environmental conditions (e.g., lower sea surface temperature, increased vertical wind shear, increased Coriolis parameter, and strengthened low-tropospheric baroclinicity) during ET [1,6–10], a subset of these systems can reintensify and produce heavy rainfall, large ocean waves, and intense gale-force winds in regions that rarely experience direct TC impacts (for example, hurricanes Floyd (1999) [11,12], Irene (2011) [13,14], and Sandy (2012) [15,16]). Moreover, the expansion of the wind field [1,6,10] and the reintensification in rare cases (e.g., highly destructive hurricane Sandy [16,17]) during ET both threaten the midlatitude region where the population density is very high.

TCs undergoing ET can produce adverse societal impacts in regions far from the original TCs. This, in conjunction with future climate changes in climatological ET regions,

motivates the investigation of the potential response of ET to climate changes [14,18–24]. Several previous studies identify an increase in the frequency or percentage of TCs undergoing ET under the representative concentration pathway (RCP) 4.5 scenario [18] or with a warmer ocean [25] in the North Atlantic and under the RCP 4.5 scenario [20] in the western North Pacific. Some studies focus on the destructive potential of TCs undergoing ET in the future. The responses of ET of Hurricane Irene (2011) to projected climate changes under the RCP 8.5 scenario are examined in Jung and Lackmann [14] (including the 2 m air temperature, sea surface temperature (SST), surface and soil temperature, and atmospheric temperature at isobaric levels) and Liu et al. [21] (including surface temperature and air temperature) by using a pseudo-global warming (PGW) approach. They both find that the storm intensity and rainfall increase in the simulations under their projected future environmental conditions. Other than case studies, Jung and Lackmann [22] use 21 North Atlantic recurving ocean ET events to initialize their simulation. Their results suggest that TCs undergoing ET could have greater potential to cause high-impact weather in Western Europe through both direct and remote processes under their projected future environmental conditions under the RCP 8.5 scenario. Michaelis and Lackmann [19,23] present an analysis using multi-seasonal global simulations representative of present-day and projected future climates with high resolution (15 km grid) throughout the Northern Hemisphere. They find increases in ET activity, intensity, and transition percentage and a northward shift in ET events in the North Atlantic under a more favorable background environment (e.g., reduced vertical wind, higher SST, higher relative humidity, and greater maximum potential intensity) in the future.

　　TC size is one of the critical physical properties for assessing the destructive potential of TCs. Many previous studies have explored the change in TC size under global warming [26–32]. They all find that TCs tend to be larger with favorable environments (e.g., reduced vertical wind, higher SST, higher relative humidity, and greater maximum potential intensity) for development in the future. Some previous studies have analyzed the expansion of the wind field associated with ET, which featured in the acceleration of the outer wind field [10,16,17]. Evans and Hart [10] analyze the physical and dynamical mechanisms behind the wind field expansion of Bonnie (1998) and attribute the acceleration of the outer wind field to the net import of absolute angular momentum along inflowing trajectories in the cyclone's environment. Shin [17] utilize the azimuthal momentum equation and 1.667 km resolution numerical simulations to study the structural changes in Hurricane Sandy's ET. Their results indicate that baroclinically driven frontal convection in the outer region induces extensive boundary layer inflow, causing a radial absolute vorticity flux, which expands Sandy's outer wind field. The expansion of the wind field enlarges the risk region of the cyclone above gale-force wind speed magnitude. The importance of assessing TCs' destructive potential increasing with global warming is already valued [33]. However, studies regarding the potential influence of climate changes on the size of cyclones undergoing ET are still few. Therefore, this study focuses on the impact of future higher SSTs on cyclone size (R17) during ET through sensitivity experiments and analyzes the physical and dynamical mechanisms behind these potential influences. The case study of Songda (2016) and a semi-idealized experiment are conducted since its lifetime maximum intensity reached 51.4 m s$^{-1}$. The ET of Songda occurred after it recurved northeastward in the western North Pacific and far away from the land, reducing the complexity caused by the impact of land. After Songda completed ET, it struck the Pacific Northwest region of the United States and Canada as a powerful extratropical cyclone, causing strong winds and heavy rain [34].

　　In Section 2, we describe the model experimental design, two methods of defining ET onset and completion, and the momentum equation for tangential wind. In Section 3, we verify our simulations and analyze the differences of the evolution of cyclone size between sensitivity experiments and the mechanisms causing these differences in the stages of TC, ET, and post-ET. Section 4 includes a summary of the results and ongoing work.

## 2. Materials and Methods

### 2.1. Model Configuration

In this study, Tropical Cyclone Songda (2016) was simulated using the two-way interactive, triply nested grid (27, 9, and 3 km) version 4.0 of the WRF Model (Figure 1a). The horizontal (x, y) domains were 301 × 247, 688 × 559, and 652 × 583. The innermost domain (3 km resolution) was the storm-following nest. In the vertical, 38 sigma levels were used with higher resolution at the bottom and upper levels, and the model top was set at 30 hPa. All domains were initialized at 0000 UTC 11 October and integrated for 66 h until 1800 UTC 13 October. D1 (27 km resolution) had a 1 h output interval, and D2 (9 km resolution) and D3 (3 km resolution) had 10 min output intervals. The NCEP 0.25°-resolution GDA/FNL analysis data (6 h interval) were used to specify the model's initial and outermost lateral boundary conditions. The model physics schemes used included (i) the WSM 6-class graupel microphysics scheme; (ii) the Eddy diffusivity Mass Flux, Quasi-Normal Scale Elimination planetary boundary layer (PBL) parameterization with the QNSE surface-layer scheme; (iii) the RRTMG scheme for longwave and shortwave radiation for all the domains, and (iv) the Kain–Fritsch (new Eta) cumulus parameterization scheme for the two outer domains (D1 and D2).

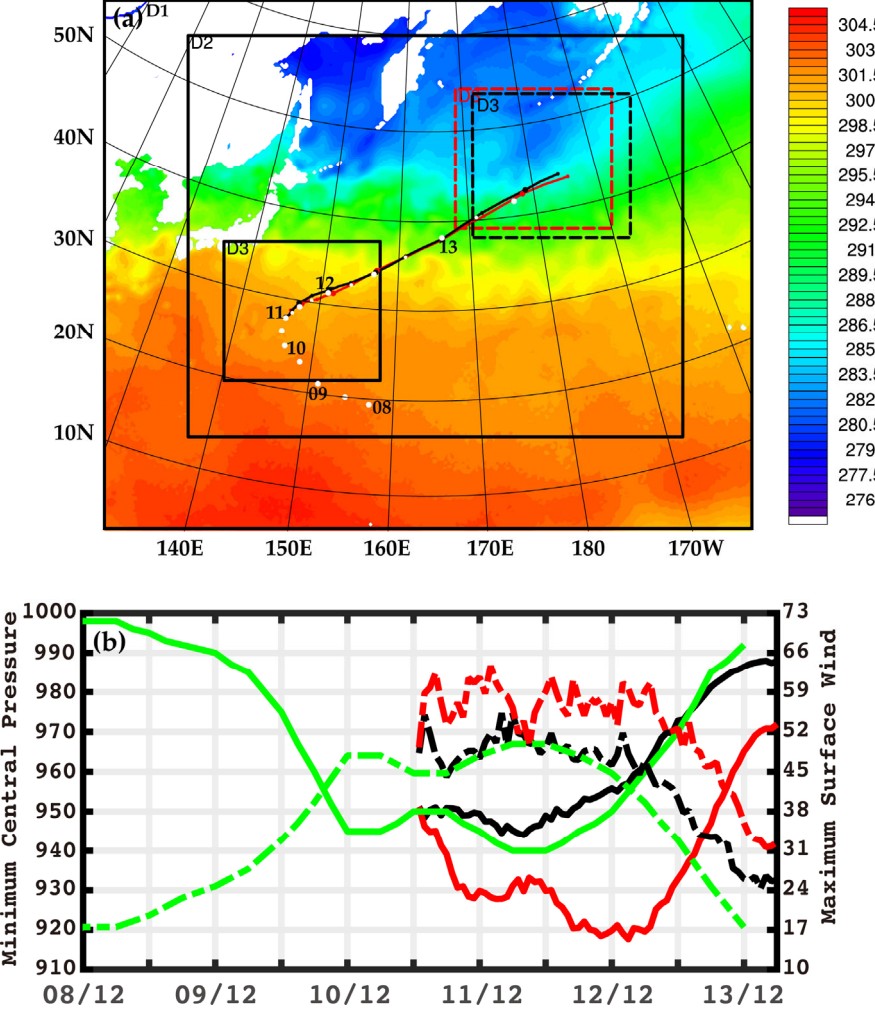

**Figure 1.** (**a**) WRF simulation domains overlaid with the SST (shaded; K) in the control at the initial time (i.e., 0000 UTC 11 October) and cyclone tracks in simulations (black solid line for the control and red solid line for SST4.5; from 0000 UTC 11 October to 1800 UTC 13 October) and best track data rest of caption (white dot; from the China Meteorological Administration (CMA); from 0000 UTC 08 October to 1200 UTC 13 October). The adjacent larger dots are 12 h intervals. D3 is the storm-following nest.

The black solid rectangle indicates the initial domains for the control and SST4.5 simulations. The black and red dashed rectangles indicate the final location of D3 in the control and SST4.5, respectively. (**b**) The minimum central pressure in simulations (black solid line for the control and red solid line for SST4.5; hPa; 1 h interval) and best track data (green solid line; hPa; 3 h interval; CMA) and maximum 10 m wind in simulations (black dashed line for the control and red dashed line for SST4.5; m s$^{-1}$; 1 h interval) and best track data (green dashed line; m s$^{-1}$; 3 h interval; CMA).

Based on the above control simulation (hereafter control), a sensitivity experiment (hereinafter referred to as SST4.5) with the SST increased by a constant value (4.5) in the entire domain was set to explore the response of the size of Songda (2016) to a projected higher sea surface temperature (SST). This constant value was obtained by referring to the pseudo global warming (PGW) method used in many previous studies [14,24,35–37]. A twenty-one multi-GCM considering the SSP5-8.5 scenario from the Coupled Model Intercomparison Project 6 (CMIP6) ensemble (as shown in Table A1) was used to project the average SST change in the future (2080–2099) relative to the historical period (1980–1999). Then, the average value of this SST change in the D2 area was calculated, corresponding to large-scale SST conditions in the late twenty-first century. The initial atmospheric conditions were the same in the SST4.5 experiment as in the control. Nakamura and Mäll [24] set four sensitivity experimental cases to consider the effect that different environmental variable changes (including the SST, atmospheric temperature, relative humidity, geopotential height, and wind velocity) have on the development of Subtropical Cyclone Anita (2010) under a future environment. The SST4.5 experiment in our study was similar to their SST-only case. The purpose was to understand separately the effect that the higher SST had on the extratropical transitioning tropical cyclone size and to propose possible mechanisms.

### 2.2. Determination of the Onset and Completion of ET

In this study, two objective methods were used to determine the onset and completion of ET. The first method was the cyclone phase space (CPS) technique [38,39], which has been widely used in research and operations. The CPS diagnosis included three parameters ($B$, $-V_T^L$, and $-V_T^U$). B denotes the difference of geopotential thickness in the lower troposphere between the right-hand and left-hand sides in the cyclone forward direction (Equation (1)). $-V_T^L$ and $-V_T^U$ describe the intensity of the thermal wind of the cyclone at the lower and upper troposphere, respectively (Equations (2) and (3)). A positive thermal wind signified a warm core, while negative values indicated a cold-core structure. All parameters were calculated within a 500 km radius from the cyclone center. In this study, the location of the minimum sea level pressure was simply defined as the center of the cyclone.

$$B = h\left[\overline{\left(Z_{600\text{hPa}} - Z_{900\text{hPa}}\right)}\bigg|_R - \overline{\left(Z_{600\text{hPa}} - Z_{900\text{hPa}}\right)}\bigg|_L\right] \tag{1}$$

$$-V_T^L = \frac{\partial \Delta Z}{\partial \ln p}\bigg|_{900\text{hpa}}^{600\text{hpa}} \tag{2}$$

$$-V_T^U = \frac{\partial \Delta Z}{\partial \ln p}\bigg|_{600\text{hpa}}^{300\text{hpa}} \tag{3}$$

Here $Z$ is the geopotential height at the corresponding pressure level; $h$ is the hemisphere-related parameter (the Northern Hemisphere is 1, and the Southern Hemisphere is $-1$); $R$ and $L$ indicate the right and left semicircles of the storm relative to storm motion, respectively; $p$ is the pressure (hPa); $\Delta Z$ equals $Z_{max} - Z_{min}$, and the vertical profiles of $\Delta Z$ are computed using pressure levels from 300 to 900 hPa at 25 hPa intervals; and the overbar denotes an average value within a 500 km radius from the cyclone center. The onset of ET is defined as the time when the values of $B$ exceed 10 m, implying significant asymmetry. The

completion of ET is defined as the time when $-V_T^L < 0$, implying the lower troposphere changes from a warm-core structure to a cold-core structure.

The CPS method was developed at a time when TC simulations were more commonly conducted with a coarse resolution. Several previous studies show that the CPS definitions brought about false alarm ET events or did not adequately reflect the ET process [3,18,20]. The CPS method may work best for datasets that do not fully resolve the intense inner warm-core TC structure. In light of these concerns, we also use an alternative method proposed by Jung and Lackmann [22]. This alternative method employs two metrics representing TC energy sources: surface latent heat flux and baroclinic conversion. The baroclinic conversion term is calculated as follows:

$$C_{pk} = \omega' \frac{R_d T'}{p} = \omega' \alpha' \tag{4}$$

Here $\omega'$ is the pressure coordinate vertical velocity, $R_d$ is the gas constant for dry air, $T'$ is the temperature deviation, $p$ is the pressure, and $\alpha'$ is the specific volume.

The energy source of transitioning storms changes from primarily diabatic to a combination of baroclinic and diabatic processes during ET. The completion of ET is defined as the time when the average of the baroclinic conversion within a 1000 km radius relative to the cyclone center exceeds the average of the surface latent heat, implying the storm's energetic evolution. This alternative method can only diagnose the completion of ET, and more work is necessary to assess its reliability and to establish the threshold values.

*2.3. Momentum Equation for Tangential Wind*

To analyze the evolution of the wind field, the momentum equation for tangential wind in cylindrical coordinates was used [17,40]. This momentum equation is

$$\frac{\partial V}{\partial t} = -\eta_a U - \frac{V}{r} \frac{\partial V}{\partial \lambda} - W \frac{\partial V}{\partial Z} - \frac{\alpha}{r} \frac{\partial p}{\partial \lambda} + 2\Omega \cos \phi \sin \lambda W + FRI \tag{5}$$

$$\eta_a = \frac{1}{r} \frac{\partial rV}{\partial r} + f \tag{6}$$

Here $U$, $V$, $W$, $\eta_a$, $p$, $\alpha$, $f$, $\phi$, and $\Omega$ are the radial velocity relative to Earth, tangential velocity relative to Earth, vertical velocity, absolute vorticity, pressure, specific volume, Coriolis parameter, latitude, and Earth angular velocity. $FRI$ is the term of friction plus the diffusion effects. Among the forcing terms on the right-hand side of the equation, the radial absolute vorticity flux ($-\eta_a U$, hereinafter referred to as RAF) plays a dominant role in generating a positive tendency of tangential wind at the lower layer [17]. To demonstrate how the frontal convection accelerates the outer wind field of Songda (2016), this study focused on the RAF at the lowest level of the WRF model.

For clear visualization, the RAF term was smoothed 10 times in space by using a weighted smoother based on a circular diffraction weighting function in the Read Interpolate Plot (RIP; a standard graphic program for WRF) program subroutine [17] and smoothed into a 1 h running mean in time.

## 3. Results
### 3.1. Verification of Simulations

According to the best track data from the China Meteorological Administration (CMA), at 1200 UTC 08 October, Songda (2016) developed into a tropical storm. The CMA upgraded Songda to a severe tropical storm at 1200 UTC 09 October, and Songda intensified rapidly later that night. At 0000 UTC 10 October, the CMA reported that Songda had developed into a typhoon. Twelve hours later (i.e., 1200 UTC 10 October), Songda became a severe typhoon. Songda maintained a severe typhoon level until 1500 UTC 12 October. After that, Songda embedded deep in the westerlies, leading to the weakening trend. At 0600 UTC 13 October, the CMA determined that Songda had completed ET. The time series of cyclone

tracks, minimum central pressure, and maximum surface wind in the control and best track data (CMA) are compared in Figure 1. The simulation results before 0600 UTC 11 October were treated as model spin-up and excluded from analyses. After 1200 UTC 12 October, the maximum surface wind speed simulated in the control experiment was slightly less than that in the best track data from CMA. The simulated lifetime minimum central pressure in the control was higher than that in the best track data from CMA. Despite some differences in the details between the control and best track data, the control experiment reproduced the TC track and the intensity evolution reasonably well. As for SST4.5, the cyclone track showed a good match with the track in the control except for some times when the latitude of the cyclone was lower than that in the control. The cyclone in SST4.5 featured greater intensity, suggesting the effect of higher SSTs in storm intensification.

### 3.2. The Onset and Completion of ET

Using the two objective methods mentioned above, the onset and completion of ET were diagnosed and are shown in Figure 2. The evolutions of the three cyclone phase space (CPS) parameters of Songda showed a good match with that of a typical ET case [38]. During ET, Songda in the control became more and more asymmetric (Figure 2c), and the thermal wind in the upper and lower troposphere gradually weakened (Figure 2a,b); the evolutions of the three CPS parameters in SST4.5 were similar to those in the control. However, the cyclone in SST4.5 featured better thermally symmetry and higher strength warm cores at the lower and upper troposphere (except for a few times) compared with that in the control. The onset and completion of ET diagnosed based on the CPS method were 1500 UTC 11 October and 1000 UTC 13 October in the control. The diagnosed onset (2100 UTC 11 October) and completion (1500 UTC 13 October) of ET in SST4.5 were both later than those in the control. However, the difference in the duration of ET (approximately 42 h) between the control and SST4.5 was very small. Based on the alternative method proposed by Jung and Lackmann [22], Songda completed ET at almost the same time (0300 UTC 13 October) in the control and SST4.5. The completion of ET (0600 UTC 13 October) determined by CMA was between the diagnosis results of the two objective methods.

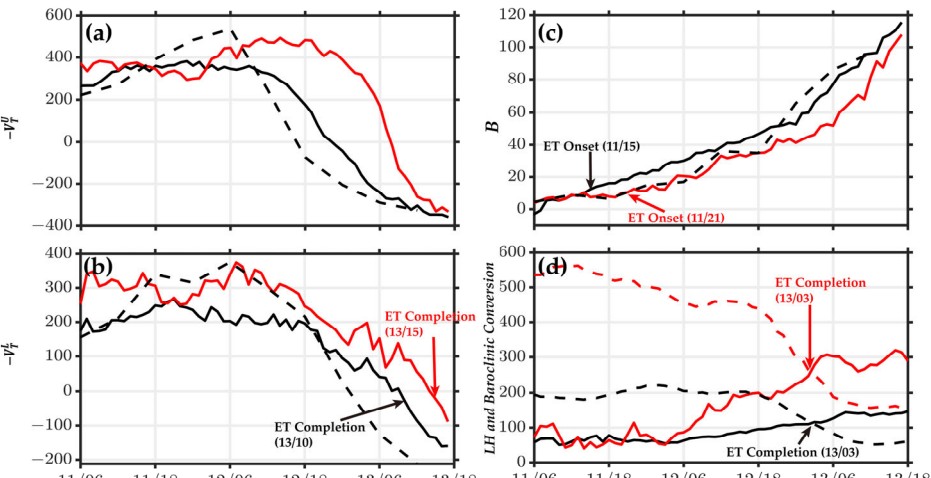

**Figure 2.** Three cyclone phase space (CPS) parameters, (**a**) $-V_T^U$, (**b**) $-V_T^L$, and (**c**) B, of Songda in simulations (black solid lines for the control and red solid lines for SST4.5) and NCEP GDA/FNL data (black dashed lines; 6 h interval). (**d**) The spatial average of the surface latent heat (dashed lines; W m$^{-2}$) and baroclinic conversion ($-\omega'\alpha'$; solid lines; W m$^{-2}$). The black lines are for the control, and the red lines are for SST4.5. The D2 output is utilized (1 h interval). The arrows indicate the onset and completion of ET diagnosed by the two methods.

### 3.3. The Impact of the Projected Higher SST on Cyclone Size

This study focuses on how a higher SST affects the cyclone size, which is simply defined as the 17 m s$^{-1}$ wind speed radius at the 10 m level (i.e., R17). On the basis of the evolution of the cyclone size in the control and SST4.5, the simulation period (except for the first 3 h) can be divided into four stages, as shown in Figure 3. During 0300-2100 UTC 11 October (stage 1; before ET or just beginning ET), the size of Songda maintained at about 165 km in the control, and the size of Songda in SST4.5 first increased from about 208 km to 234 km and then maintained at about 225 km. During 2100 UTC 11 October to 0700 UTC 12 October (stage 2), Songda had begun ET, and the cyclone outer wind fields accelerated earlier in the control (stage 2a) and later in SST4.5 (stage 2b). The cyclone size in the control first increased from about 165 km to about 215 km (about 50 km in all) and then maintained. The cyclone size in SST4.5 began to increase at 0700 UTC 12 October, which was 10 h later than that in the control (2100 UTC 11 October), and increased about 56 km within the next two hours and then fluctuated around 250 km. During 1800 UTC 12 October to 0600 UTC 13 October (stage 3), Songda was about to complete ET. The storm weakened, and the cyclone size reduced by about 100 km in the control and about 50 km in SST4.5. During 0600-1800 UTC 13 October (stage 4), Songda had basically evolved into a frontal cyclone, and the cyclone size increased again by about 85 km in the control and about 120 km in SST4.5.

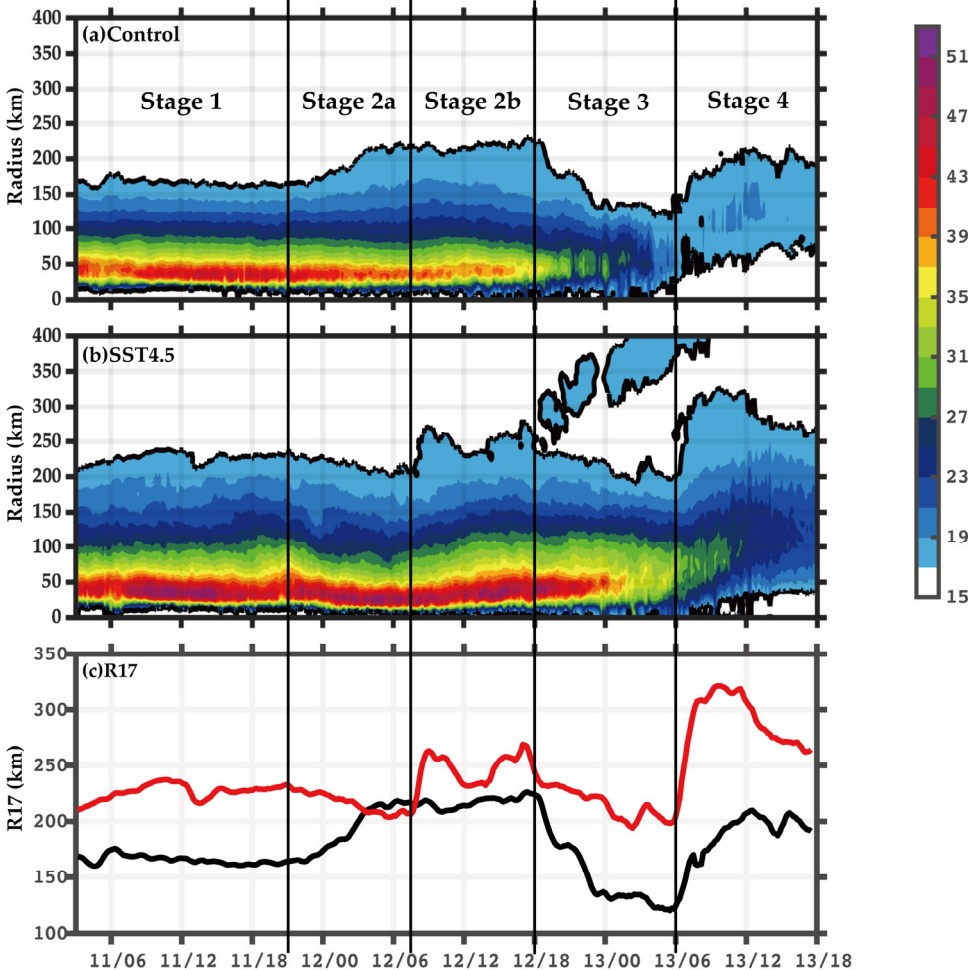

**Figure 3.** Horizontal wind speed (shaded; m s$^{-1}$) at the 10 m level in (**a**) the control and (**b**) SST4.5. The vertical axes show the distance (km) from the storm center. (**c**) Radius of the 10 m wind speed 17 m s$^{-1}$ (R17; km; 1 h running mean) in the control (black solid lines) and SST4.5 (red solid lines). and SST4.5 (red solid lines). The D3 output is utilized (10 min interval). Different stages of cyclone

size evolution are separated by thin vertical lines. Stage 1 was before ET or just beginning ET; in stage 2, the cyclone size increased earlier in the control (stage 2a) and later in SST4.5 (stage 2b); in stage 3, the cyclone size reduced in both the control and SST4.5; and in stage 4, the cyclone size increased again in both the control and SST4.5.

In summary, Songda experienced the expansion of outer wind field during ET in both the control and SST4.5, which usually occurs in typical ET events. There were two major differences in the evolution of cyclone size between the control and SST4.5. First, the cyclone size in SST4.5 was larger than that in the control (except for a few times), enlarging the risk region of the cyclone above gale-force wind speed magnitude. In addition, the cyclone size increasing in SST4.5 lagged behind that in the control, changing the affected region and timing by cyclone gale-force winds.

*3.4. The Mechanism for Variations of Cyclone Size (R17) Evolution with Different SST Scenarios*

To explain the differences in the cyclone size evolution between the control and SST4.5, as mentioned in the previous section, the azimuthally averaged RAF term, which plays a dominant role in generating a positive tendency of tangential wind at the model's lowest level, is displayed in Figure 4. Figure 4 shows that the evolution of the positive RAF in the outer region and the evolution of the cyclone size match well, implying that it is the positive RAF near the location of cyclone R17 that dominates the evolution of the cyclone size. Considering that the mechanism of the projected higher SST affecting cyclone size may be different before and after the interaction between Songda and the midlatitude baroclinic zone, the aforementioned four stages will be analyzed separately in the following sections.

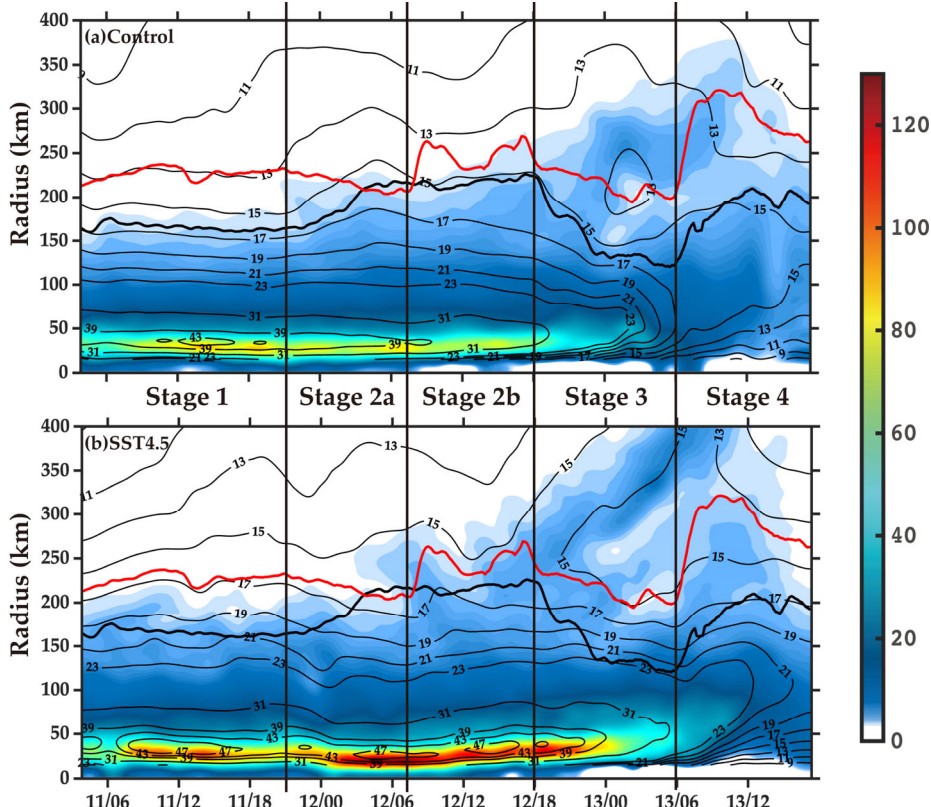

**Figure 4.** (**a**) The azimuthally averaged radial absolute vorticity flux (RAF; shaded; $> 3$ m s$^{-1}$h$^{-1}$), azimuthally averaged tangential wind (black contours; m s$^{-1}$) in the control, and the cyclone R17 in the control (black solid lines; km) and SST4.5 (red solid lines; km). The vertical axes show the distance (km) from the storm center. (**b**) As in (**a**) but for SST4.5. The D3 output is utilized (10 min interval). Different stages of the cyclone R17 evolution are as in Figure 3. For clear visualization, the RAF term and tangential wind are smoothed. Contours within 15 km are not displayed.

### 3.4.1. Before ET (Stage 1)

Figure 5 shows the column maximum reflectivity in the control and SST4.5 and the radial-vertical cross-section of the differences of reflectivity, radial velocity, and vertical velocity between the control and SST4.5 at 0600 UTC 11 October. This period was chosen to represent stage 1 (i.e., before ET or just beginning ET). During stage 1, the interaction between Songda and the baroclinic zone was weak. Songda still maintained the features of a mature TC. Therefore, the projected higher SST mainly affected the cyclone size by affecting the TC itself. Tsuji et al. [41] used an idealized model to simulate the relationship between rainbands and TC size, defined as the radius of the wind speed of 15 m s$^{-1}$ at 10 m height (i.e., R15). They find that R15 depends on the location of thermal forcing related to rainbands. When the forcing is applied to the outer part of a vortex but still inside R15, the secondary circulation extends to the outside of R15 and carries absolute angular momentum (AAM) inward. Thus, R15 increases. Conversely, when the forcing is applied near the center of the vortex, where inertial stability is strong, the secondary circulation closes inside R15, and R15 hardly increases.

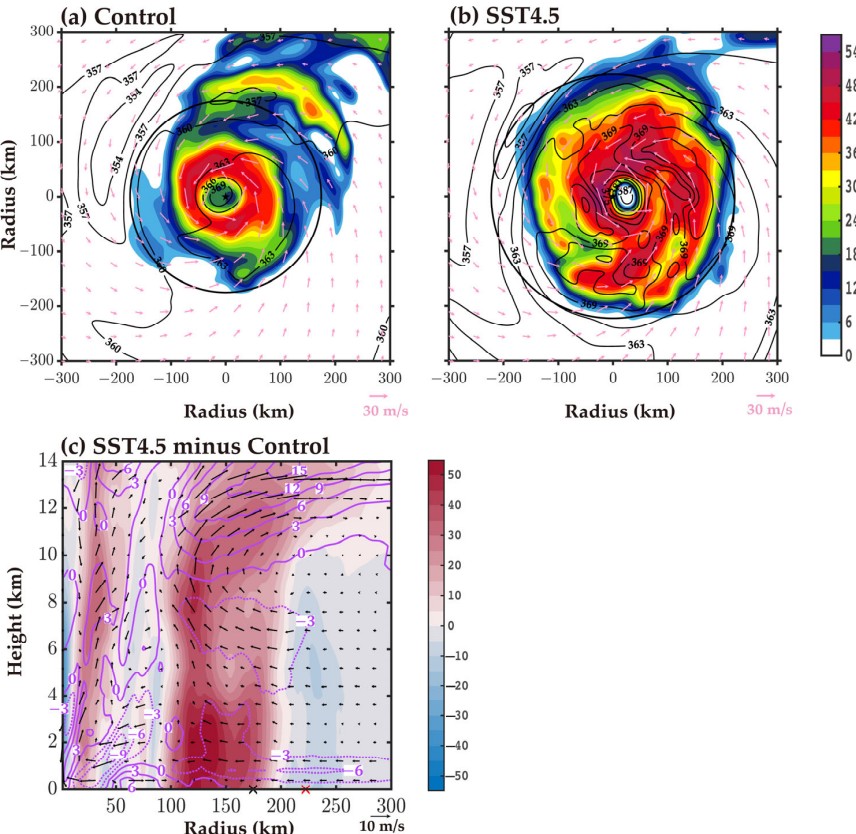

**Figure 5.** (**a**) The column maximum reflectivity (shaded; dBZ) and equivalent potential temperature at the model's lowest level (black contours; 3 K interval) in the control at 11/06. Horizontal flow vectors are superimposed. For clear visualization, reflectivity and equivalent potential temperature are smoothed. The horizontal and vertical axes indicate the distance (km) from the storm center (black pentagram). (**b**) As in (**a**) but for SST4.5. (**c**) The difference of azimuthally averaged reflectivity (shaded; dBZ) and radial velocity (purple contours; 3 m s$^{-1}$ interval; negative values are dotted for inflow, and positive values are solid for outflow) between SST4.5 minus the control at 11/06. The differences of the in-plane flow vectors (vertical velocity component multiplied by 5) are superimposed. The black and red crosses indicate the cyclone R17 in the control and SST4.5 at 11/06, respectively. The horizontal and vertical axes indicate the distance (km) from the storm center and height (km), respectively. In (**c**), the horizontal and vertical axis indicate the distance (km) from the storm center and height (km), respectively. The D3 output at 0600 UTC 11 October is utilized for (**a**–**c**).

Figure 2d shows that the projected higher SST enhanced surface heat fluxes over the ocean in the entire domain. According to Xu et al. [29], more surface heat fluxes lead to more active rainbands in both the inner-core (near 25 km) and outer-core (near 125 km) regions (Figure 5a–c), resulting in an increase in diabatic heat release. As a result, the secondary circulation including the low-level inflow (Figure 5c) near R17 is enhanced and extends outward. In Figure 4, strong RAF ($> 3 \text{ m s}^{-1}\text{h}^{-1}$) in SST4.5 extends outward about 30 km compared with that in the control, which is similar to the role of radial advection of absolute angular momentum (AAM) per unit radius. As a result, the R17 in SST4.5 is larger.

### 3.4.2. During ET (Stage 2)

Shin [17] has revealed that peripheral baroclinically driven frontal convection induces extensive boundary layer inflow, leading to strong inward AAM transport, which accelerates the horizontal flow in the outer region and expands the cyclone's outer wind field during ET. To understand how the projected higher SST affects the cyclone size evolution by affecting the frontal convection resulting from the interaction between TC and the baroclinic zone, the horizontal structure of the radar reflectivity, potential temperature, and inward radial velocity at the model's lowest level at 2100 UTC 11 October, 0200 UTC 12 October, and 0600 UTC 12 October are shown in Figure 6. Figure 6 shows that strong radar reflectivity and radial velocity occurred in the north of Songda during ET, implying that Songda gradually embedded and interacted with the baroclinic zone. At 2100 UTC 11 October (Figure 6a,d), the latitude of Songda in SST4.5 (red pentagram) was lower than that in the control (black pentagram). However, the difference in the location of the midlatitude baroclinic zone (green crosses) between the control and SST4.5 was not significant. As a result, the distance between Songda and the baroclinic zone in SST4.5 was farther than that in the control, leading to a weaker interaction between them, which featured weaker reflectivity and weaker inflow in the north of Songda. At 0200 UTC 12 October (Figure 6b,e), the strong reflectivity and inflow driven by frontal convection in the north of Songda approached the cyclone center gradually. The frontal convection in the control had reached the position of the cyclone R17 (about 180 km), but the frontal convection in SST4.5 was still far from the cyclone center (about 300 km). As will be shown later, the broad, intense inflow driven by frontal convection will accelerate the outer wind field of the cyclone through strong inward advection of AAM per unit radius (i.e., RAF). Therefore, the variation of the location of frontal convection relative to the cyclone during ET will result in the variation of the cyclone size evolution.

Usually, the steering flow can affect the tropical cyclone track [42]. Figure 7 shows the relationship between the meridional steering flow and the latitude of Songda. As expected, when the difference of meridional steering flow (the north direction is defined as positive) between the control and SST4.5 was negative, the difference of the latitude of the cyclone (the north direction is defined as the positive direction) between the control and SST4.5 decreased. Conversely, when the difference of the meridional steering flow between the control and SST4.5 was positive, the difference of the latitude of the cyclone between the control and SST4.5 increased. The latitude of Songda in SST4.5 was lower than that in the control during 1500 UTC 11 October-0600 UTC 12 October (about 0.35° at most).

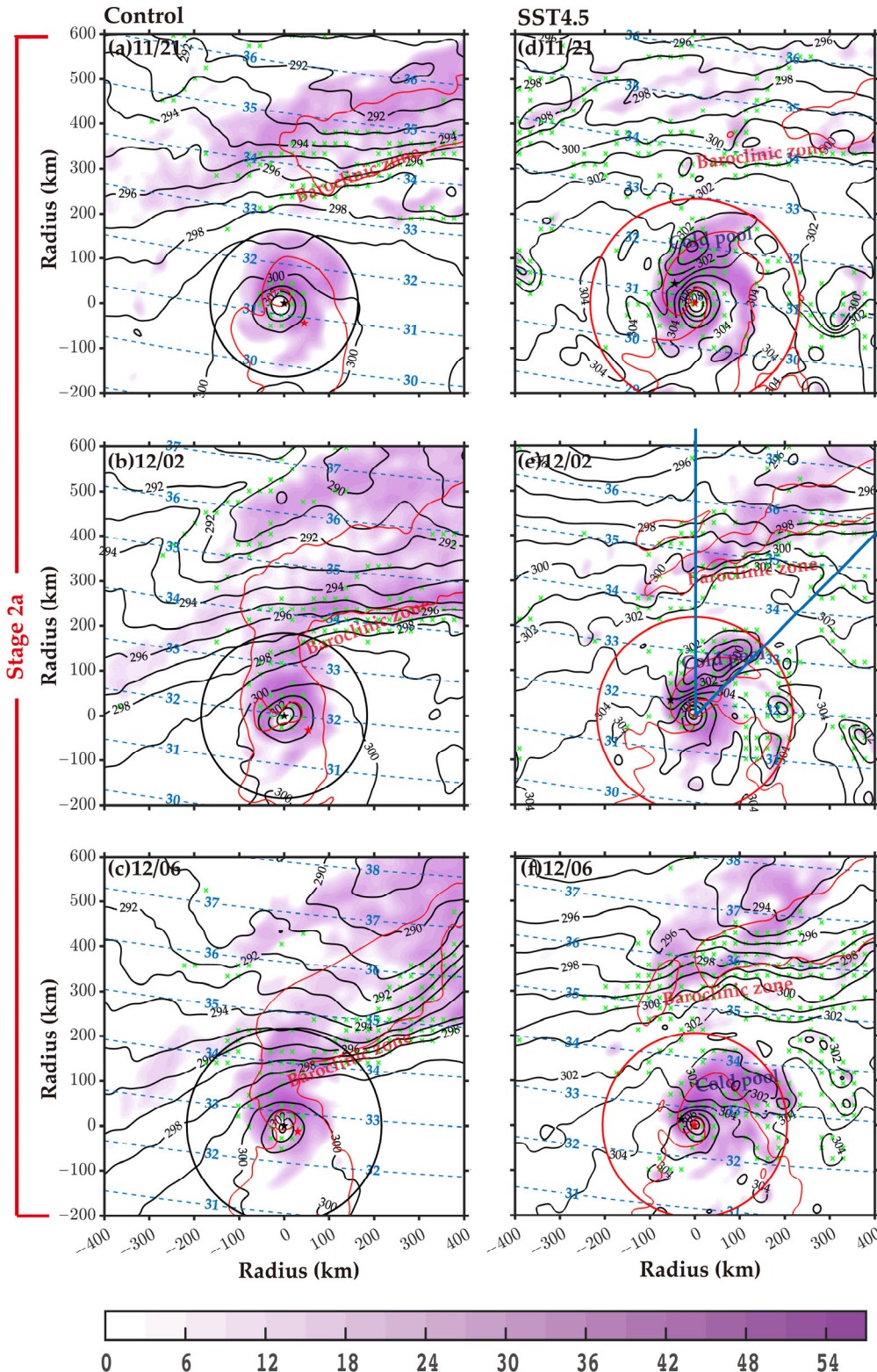

**Figure 6.** (**a**–**c**) Horizontal structure of reflectivity (shaded; dBZ), potential temperature (black contours; 1 K interval), and inward radial velocity (red contour at −12 m s$^{-1}$) at the model's lowest level at 11/21, 12/02, and 12/06 in the control, respectively. (**d**–**f**) As in (**a**–**c**) but for SST4.5. The green crosses indicate the approximate region where the potential temperature gradient is above 3 K (100 km)$^{-1}$. The horizontal and vertical axis indicate the distance (km) from the storm center (black pentagram for the control and red pentagram for SST4.5). The black circle and red circle indicate cyclone R17 in the control and SST4.5, respectively. The dark blue dashed contours indicate latitudes.

The words "Baroclinic Zone" and "Cold pool" indicate their approximate locations. Two dark blue lines in (**e**) indicate the segment over which the variables are azimuthally averaged in Figure 8. For clear visualization, the reflectivity, potential temperature and radial velocity are smoothed. The D3 output is utilized.

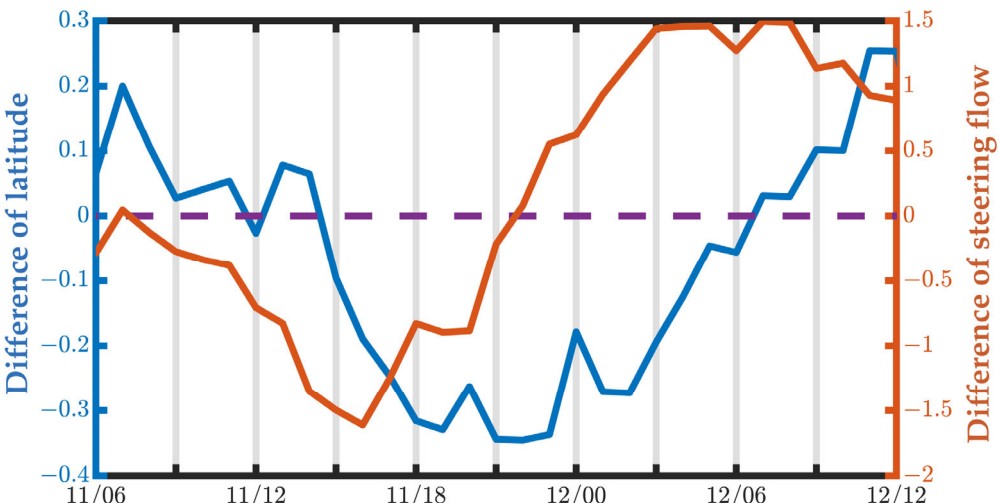

**Figure 7.** The differences of the central latitude of Songda (dark blue lines; degree) and meridional steering flow (orange lines; ) between the control and SST4.5. The steering flow is defined as the horizontal wind vertically averaged below 200 hPa with mass weighting after taking the horizontal average within a 500 km radius from the cyclone center (c.f. the definition in [42]). The dotted purple line is the zero reference line. The D2 output is utilized.

Furthermore, at 0600 UTC 12 October (Figure 6c,f), although the difference of the latitude of Songda between the control and SST4.5 was very small, the distance between the frontal convective region and the cyclone center in SST4.5 was still farther than that in the control. We note that with more active rainbands in the northern part of the cyclone (Figure 6d–f), a strong surface cold pool forms beneath these precipitation clouds during stage 2a in SST4.5 because the evaporative cooling-driven downdrafts bring cold and dry air from the middle troposphere to the boundary layer [43]. It needs to be examined if this cold pool affects the interaction between Songda and the baroclinic zone. The azimuthally averaged low-level scalar frontogenesis function, potential temperature departure, and radial velocity are displayed in Figure 8. Figure 8c shows that at 0200 UTC 12 October (stage 2a), a cold center of potential temperature (i.e., cold pool) existed in the north of the cyclone at a lower layer in SST4.5. Strong frontolysis existed, and the inflow was weak outside of it. This was not conducive to the peripheral frontal convection approaching the cyclone center. As for the control (Figure 8a), continuous frontogenesis and intense inflow existed at the lower layer of Songda without a cold pool. The lower latitude of the cyclone and the effect of the cold pool jointly led to the farther distance between the cyclone and the frontal convective region in SST4.5. As a result, the peripheral frontal convection reached the location of the cyclone R17 later in SST4.5 so that the increase in the cyclone size driven by frontal convection in SST4.5 was later than that in the control.

To examine the role of the radial absolute vorticity flux (i.e., RAF) in accelerating the outer tangential wind field of Songda, the tangential velocity and RAF at the model's lowest level at 2100 UTC 11 October, 0200 UTC 12 October, and 0600 UTC 12 October in the control and SST4.5 are shown in Figure 9. Only positive RAF near R17 can cause R17 to increase. Figures 6 and 9 show that the region of the intense positive RAF near R17 of cyclone was almost in agreement with the region of the frontal convection induced boundary layer inflow, which implied that it was the peripheral frontal convection rather than the convection in TC core region that caused intense inward RAF in outer region, which played

a dominant role in accelerating the outer wind field of Songda (2016) during ET. In the control, the region of positive RAF driven by frontal convection is close to the position of the cyclone R17 at 2100 UTC 11 October (Figure 9a). At 0200 UTC 12 October (Figure 9b) and 0600 UTC 12 October (Figure 9c), the frontal convection region entered the range of the cyclone R17. The low-level outer tangential flow of Songda accelerated in the inflow region due to positive RAF, which extended intense tangential flow to the downstream side of the frontal convection. As a result, the tangential flow in the northwestern quadrant accelerated (Figure 9a–c), and the cyclone size increased in this stage (Figure 3a,c). As for SST4.5, because Songda is always far from the frontal convection during stage 2a, the strong RAF driven by frontal convection was located outside of the cyclone R17 (Figure 9d–f). The weak RAF near the cyclone R17 was not enough to offset the effects of other negative contributions. In addition, the negative RAF driven by the weak outflow existed at the outside of the cold pool. Moreover, Tsuji et al. [41] found that thermal forcing related to rainbands located at the outside of R15 could result in the decrease in R15. As a result, the cyclone R17 reduced during stage 2a in SST4.5.

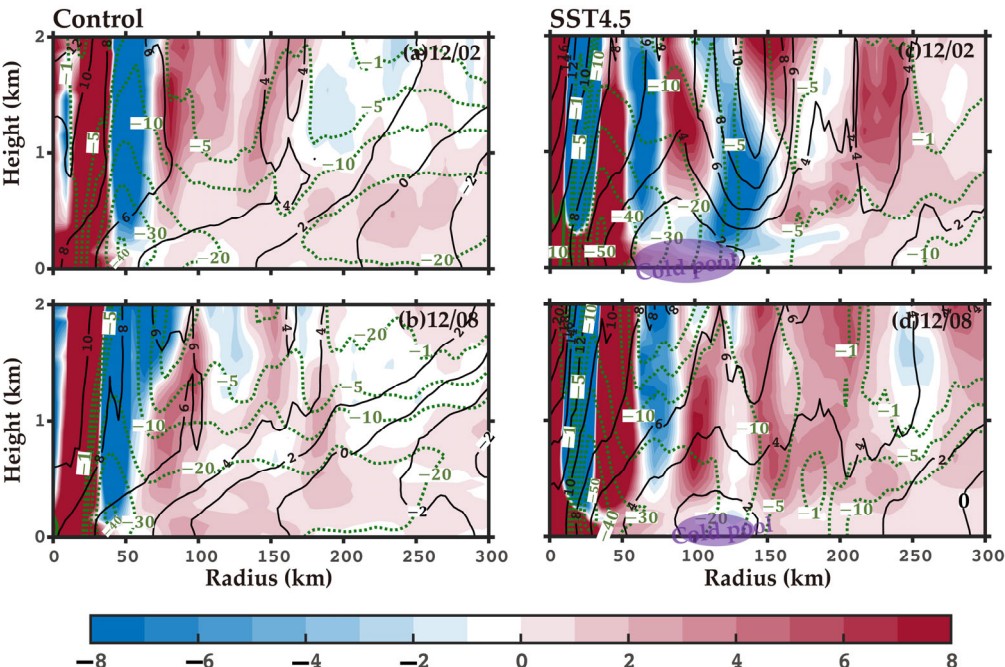

**Figure 8.** Radius–height cross-section of potential temperature departure (black contours; 2 K interval; from D3 output), radial velocity (green contours; m s$^{-1}$; positive values are solid for outflow, and negative values are dotted for inflow; from D3 output), and scalar frontogenesis function (shaded; $10^{-8}$ K m$^{-1}$ s$^{-1}$; positive values indicate frontogenesis, and negative values indicate frontolysis; the D2 output is utilized) at (**a**) 12/02 and (**b**) 12/08 in the control. (**c,d**) As in (**a,b**) but for SST4.5. The variables are azimuthally averaged over the segments that are indicated by two dark blue lines in Figures 6e and 10c. The purple words "cold pool" and shadows indicate their approximate locations; 12/02 is chosen to represent stage 2a, and 12/08 is chosen to represent stage 2b.

In the same way as stage 2a, the evolution of the cyclone size during stage 2b was analyzed (Figures 10 and 11). At 0800 UTC 12 October (Figure 10a) and 1300 UTC 12 October (Figure 10b), the frontal convection intruded into the cyclone center and continued to generate strong positive RAF in the control (Figure 11a,b). Therefore, the cyclone R17 maintained (Figure 3c) in this stage. As for SST4.5, the low-level cold pool gradually weakened and disappeared at 0800 UTC 12 October and 1300 UTC 12 October (Figure 10c,d). As a result, the frontolysis at the outside of the cold pool weakened (Figure 8d), and the frontal convection reached the position of the cyclone R17 (Figure 10c,d). As stage 2a in the control, strong RAF (Figure 11c,d) driven by the frontal convection accelerated the low-level outer tangential flow

near the cyclone R17. In addition, the frontal convection in the south of the cyclone accelerated the tangential flow in the southeastern quadrant through strong RAF. As a result, the cyclone size in SST4.5 first increased and then maintained (Figure 3c) during stage 2b.

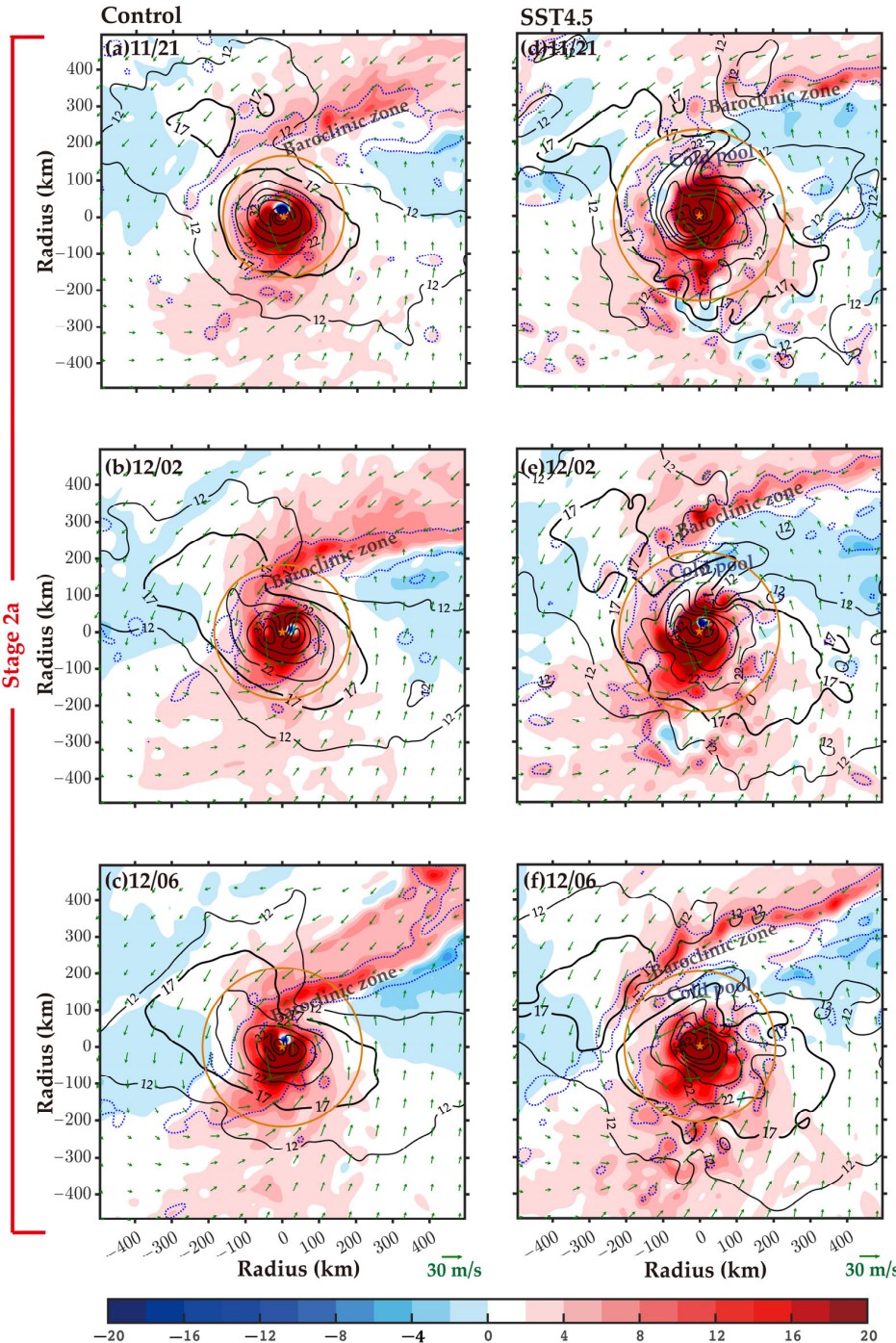

**Figure 9.** (**a**–**c**) Horizontal structure of radial absolute vorticity flux (RAF; shaded; m s$^{-1}$ h$^{-1}$), tangential velocity (black contours; 5 m s$^{-1}$ interval), absolute vorticity (blue dotted contour at $15 \times 10^{-5}$ s$^{-1}$), and horizontal flow vectors (green arrows) at the model's lowest level at 11/21, 12/02, and 12/06 in the control, respectively. The horizontal and vertical axes indicate the distance (km) from the storm center (orange pentagram). The orange circles indicate the R17 of the cyclone. (**d**–**f**) As in (**a**–**c**) but for SST4.5. The words "Baroclinic Zone" and "Cold pool" indicate their approximate locations. For clear visualization, the RAF, tangential velocity, and absolute vorticity are smoothed. The D3 output is utilized.

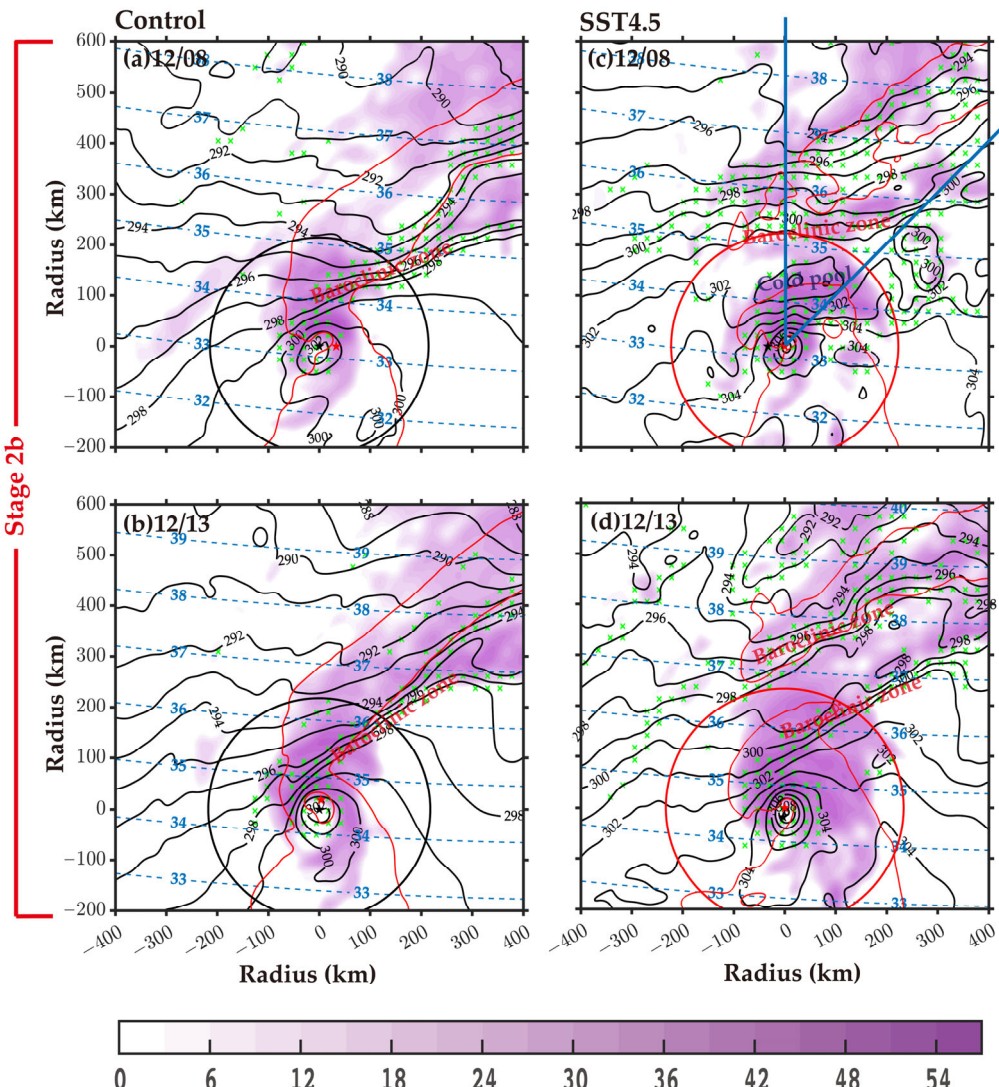

**Figure 10.** As in Figure 6 but for 12/08 and 12/13 (stage 2b). Two dark blue lines in **c** indicate the segment over which the variables are azimuthally averaged in Figure 8.

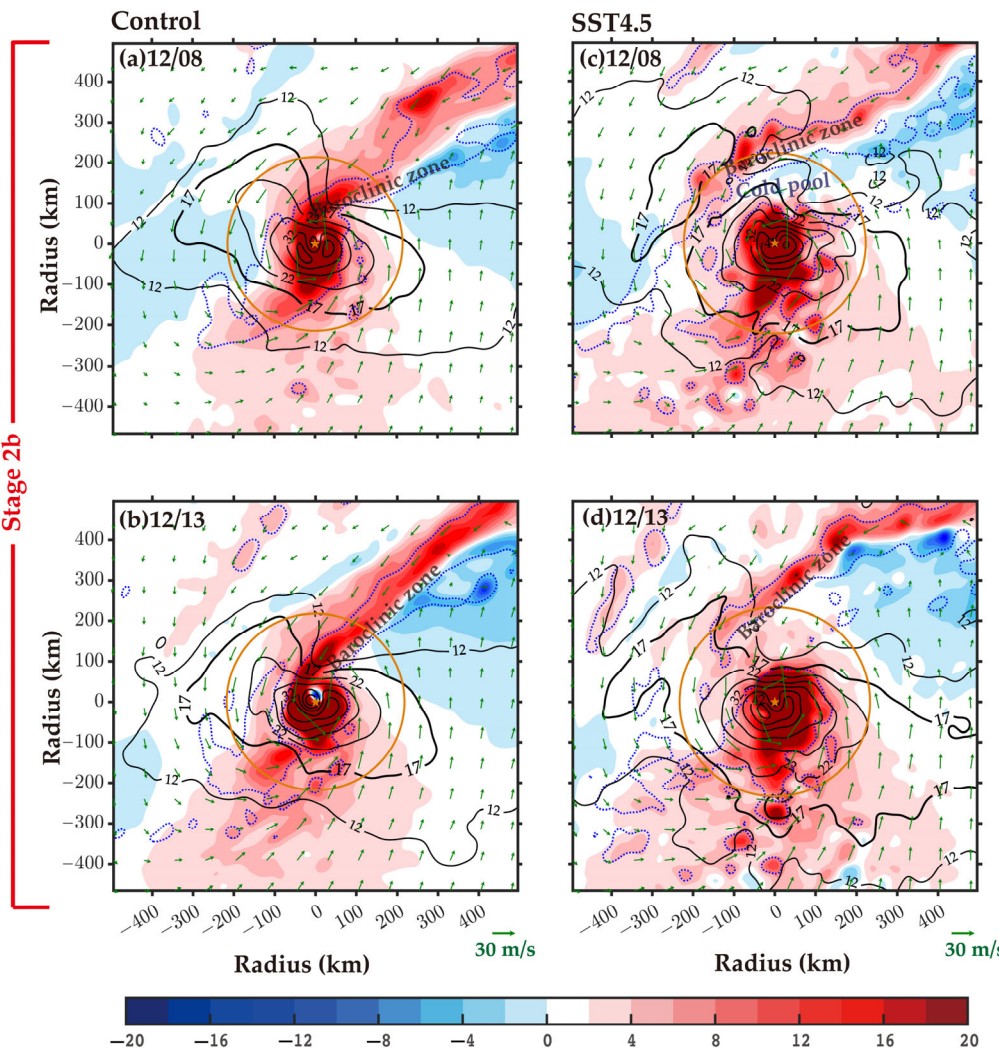

**Figure 11.** As in Figure 9 but for 12/08 and 12/13 (stage 2b).

### 3.4.3. About to Complete ET (Stage 3) and after ET (Stage 4)

During stage 3 and stage 4, Songda basically transformed into a frontal cyclone. During stage 3, the previous frontal convection region was detached from the core region and propagated to the outer region, and another frontal convection developed in the northeastern quadrant in the control and in SST4.5 (Figure 12). However, it was still far from the center of cyclone. As a result, without strong RAF (Figure 13) driven by frontal convection near the position of the cyclone R17, the cyclone size reduced (Figure 3c). During stage 4, the new frontal convection (Figure 12) approached the center of Songda, causing the cyclone size to increase again through strong RAF (Figure 3c).

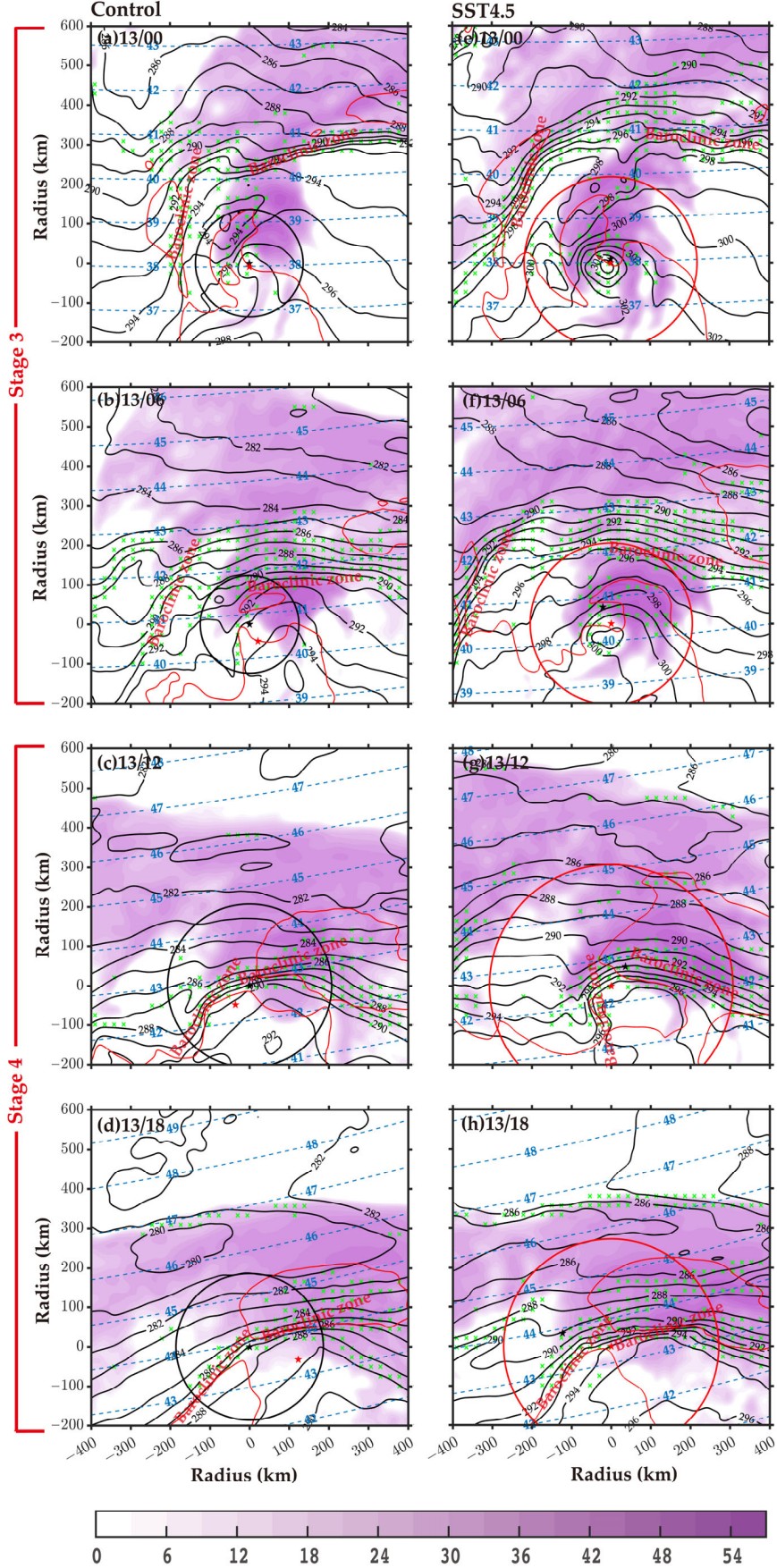

**Figure 12.** As in Figure 6 but for 13/00, 13/06, 13/12, and 13/18 (stage 3 and stage 4).

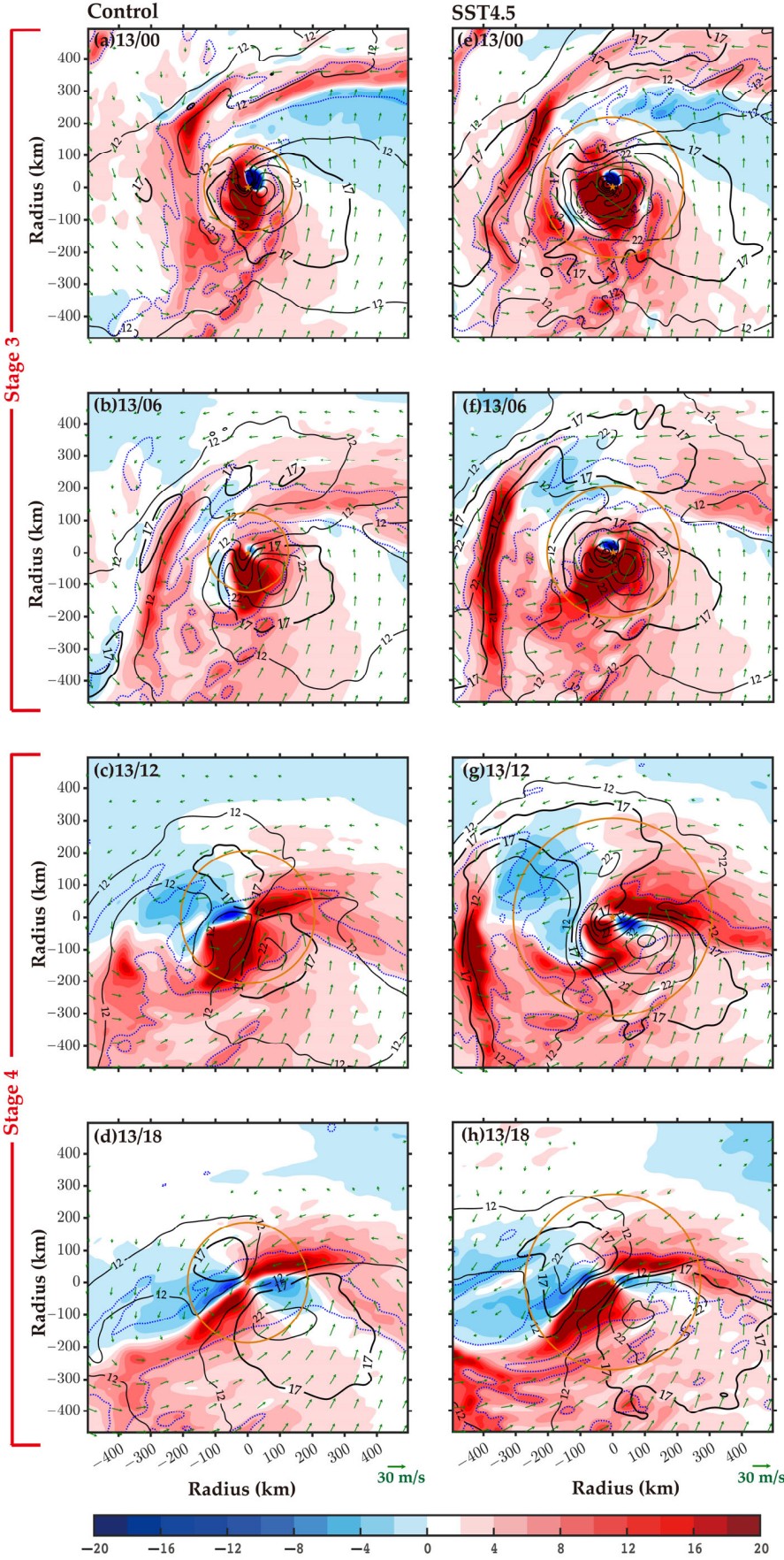

**Figure 13.** As in Figure 9 but for 13/00, 13/06, 13/12, and 13/18 (stage 3 and stage 4).

## 4. Discussion and Conclusions

This study explored the response of the cyclone size (R17) of Songda (2016) to the projected higher SST in the stages of TC, ET, and post-ET through an SST sensitivity experiment. We mainly focused on the mechanism during ET. The results showed two major differences in the evolution of cyclone size between the control and SST4.5 scenarios: First, the cyclone size in SST4.5 was larger than that in the control. Second, the increase in the cyclone size during ET in SST4.5 lagged behind that in the control. These two differences involved the affected region and the timing of cyclone gale-force winds. The mechanisms for the variations of the cyclone size (R17) evolution with different SST scenarios differed before and after the interaction between Songda and the midlatitude baroclinic zone.

Before ET, Songda still maintained the features of mature TCs. The interaction between Songda and the midlatitude baroclinic zone was weak. The projected higher SST mainly affected the cyclone R17 by affecting the TC itself. With increased SSTs in the entire domain, the surface heat fluxes over the ocean increased, which was consistent with many prior studies [27,29]. This was conducive to the development of rainbands in both the inner-core and outer-core regions, leading to an increase in the diabatic heat release in the rainbands. As a result, the stronger secondary circulation and low-level inflow extended outward, and the absolute angular momentum importing from the outer region increased, which accelerated the tangential wind of cyclone and led to a larger R17. Chavas et al. [44] analyzed the statistical distribution of two size metrics (the radius of 12 m s$^{-1}$ (R12) and the outer radius of vanishing wind (R0)), as well as their dependence on the relative SST, defined as the difference between the local SST and its tropical mean value using observation data. They found that the mean storm size increased systematically with the relative SST. The dependence of R12 on the relative SST was approximately 48 km/K. In our study, with SSTs increased by 4.5 K in the entire domain, the R17 of cyclone increased by about 60 km before ET. The dependence of R17 on a higher SST was approximately 13.3 km/K, which was comparable to, although smaller than, 48 km/K.

During ET, as Songda embedded deeper into the baroclinic zone, the baroclinically driven frontal convection region developed in the northeast quadrant owing to confluence in both the control and SST4.5 experiments. In response to frontogenesis, a broad, intense inflow driven by frontal convection occurred near the frontal convection region and accelerated the outer wind field of Songda through strong radial absolute vorticity flux (i.e., RAF). When this peripheral frontal convection reached the position of the cyclone R17, the cyclone size increased. However, during the early period of ET (stage 2a), the latitude of Songda in SST4.5 was lower than that in the control due to the smaller meridional steering flow. Moreover, one strong cold pool existed about 100 km to the north of Songda due to evaporative cooling in the outer-core rainband region. A strong frontolysis existed, and the inflow was weak outside of it, which hindered the interaction between Songda and the baroclinic zone. The lower latitude of Songda and the effect of the cold pool in SST4.5 jointly led to a farther distance between Songda and the baroclinic zone, causing the peripheral frontal convective region and the intense RAF to reach the location of cyclone R17 later. As a result, the increase in cyclone size during ET in SST4.5 was later than that in the control.

In short, it was the more active convection in the TC core region causing a larger R17 in SST4.5 compared with the control before ET. It was the intense RAF owing to frontal convection in the outer region accelerating the outer wind field and leading to R17 increasing during ET in both the control and SST4.5.

After ET, in both the control and SST4.5, the previous frontal convection region was detached from the core region and propagated to the outer region. A new frontal convection then developed in the northeastern quadrant and gradually approached the center of Songda. As a result, the cyclone size first reduced and then increased again.

In conclusion, cyclones that undergo ET can tend to be stronger and larger in all the stages of TC, ET, and post-ET and can potentially pose a greater threat of gale-force wind to the regions that rarely experience direct TC impacts with higher SSTs in the future. In

addition, the timing of the wind field expansion during ET may change, which also changes the affected region and timing of cyclone gale-force winds. Before ET, the projected higher SST affects the evolution of cyclone size mainly by affecting the cyclone itself. During ET, the projected higher SST can also affect the evolution of cyclone size by affecting the interaction between the transitioning cyclone and the midlatitude baroclinic zone.

There are several limitations to our current study. Given that only the effect of the projected higher SST is considered, these modeling results should be further improved to consider the temperature changes in the oceanic mixed layer and the cooling effect that strong wind speeds have on the ocean surface. Moreover, Nakamura and Mäll (2021) found that the intensity of Cyclone Anita was weaker in cases with future air temperature changes considered due to high static stability fields and reduced latent heat release compared with the SST-only case [24]. Future studies of more environmental variable changes, such as atmospheric temperature, relative humidity, geopotential height, and wind velocity, are required. Moreover, the effects of the baroclinity of the baroclinic zone, the upper-level trough, and the jet stream on cyclone size are absent in the present study. More case studies are also needed to verify the universality of these results.

**Author Contributions:** Conceptualization, X.T.; methodology, Z.M. and X.T.; software, Z.M.; validation, Z.M.; formal analysis, Z.M.; investigation, Z.M.; resources, X.T.; data curation, Z.M.; writing—original draft preparation, Z.M.; writing—review and editing, X.T.; visualization, Z.M.; supervision, X.T.; project administration, X.T.; funding acquisition, X.T. All authors have read and agreed to the published version of the manuscript.

**Funding:** This work was funded by the National Natural Science Foundation of China (grants 42275049 and 42192555).

**Institutional Review Board Statement:** Not applicable.

**Informed Consent Statement:** Not applicable.

**Data Availability Statement:** CMIP6 data were downloaded online (https://esgf-node.llnl.gov/search/cmip6/, accessed on 13 December 2021). NCEP and NCAR provided the GDAS/FNL 0.25 degrees global tropospheric analyses and forecast grid data. The GDAS/FNL data were accessed on 5 January 2022. The best track data for Songda (2016) were download online (https://www.ncei.noaa.gov/data/international-best-track-archive-for-climate-stewardship-ibtracs/v04r00/access/, accessed on 24 December 2021).

**Acknowledgments:** The numerical calculations in this paper were performed at the computing facilities in the High Performance Computing Center of Nanjing University and on TianHe-1 (A) at the National Supercomputer Center in Tianjin. We thank four reviewers for their helpful comments.

**Conflicts of Interest:** The authors declare no conflict of interest.

## Appendix A

**Table A1.** Information about the Coupled Model Intercomparison Project Phase 6 (CMIP6) models used in this study (some of the models similar to those selected in Dong et al. [45]; available from https://esgf-node.llnl.gov/search/cmip6/; accessed on 13 December 2021).

| Source ID | Institution ID | Nominal Resolution (km) |
| --- | --- | --- |
| ACCESS-CM2 | CSIRO-ARCCSS | 250 |
| ACCESS-ESM1-5 | CSIRO | 250 |
| BCC-CSM2-MR | BCC | 100 |
| CAMS-CSM1-0 | CAMS | 100 |
| CanESM5 | CCCma | 100 |
| CESM2 | NCAR | 100 |
| CESM2-WACCM | NCAR | 100 |
| CNRM-CM6-1 | CNRM-CM6.1 | 100 |

**Table A1.** *Cont.*

| Source ID | Institution ID | Nominal Resolution (km) |
|---|---|---|
| CNRM-CM6-1-HR | CNRM-CM6.1 | 25 |
| CNRM-ESM2-1 | CNRM-ESM2.1 | 100 |
| EC-Earth3-Veg | EC-Earth-Consortium | 100 |
| FGOALS-f3-L | CAS | 100 |
| GFDL-CM4 | NOAA-GFDL | 25 |
| GFDL-ESM4 | NOAA-GFDL | 50 |
| IPSL-CM6A-LR | IPSL | 100 |
| MIROC-ES2L | MIROC | 100 |
| MIROC6 | MIROC | 100 |
| MRI-ESM2-0 | MRI | 100 |
| NESM3 | NUIST | 100 |
| NorESM2-LM | NCC | 100 |
| UKESM1-0-LL | MOHC | 100 |

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
