# Peer review of "Response of Extratropical Transitioning Tropical Cyclone Size to Ocean Warming: A Case Study for Typhoon Songda in 2016"

_atmosphere, doi:10.3390/atmos14040639_

Round 1

Reviewer 1 Report

Title: Response of Extratropical Transitioning Tropical Cyclone Size to Ocean Warming

Authors: Miao et al.

Summary:

This manuscript analyzed the potential change of TC size for TCs undergoing extratropical transitioning (ET) in a projected ocean warming scenario using WRF simulations. Possible mechanisms are proposed to explain the observed difference of TC size change.

Though studying the TC intensity change or other relevant properties under a warming ocean scenario is not new, this study is so thorough that it is worth being published.

This manuscript is well-written. The presentation is comprehensive and thorough. The figures are of high quality and easy to follow. It is quite rare to see such a comprehensive study in the journal of Atmosphere.

I have no severe concerns about this manuscript, and it should be published after a major revision.

My main concern is that the current title is too broad. The authors might want to indicate this is a case study of Typhoon Songda. Besides, I have a few technical questions, most related to your methodology section, which should be easy to handle.

Recommendation:

Major revision

Major comments:

1.     I strongly suggest that the authors should revise their title as “Response of Extratropical Transitioning Tropical Cyclone Size to Ocean Warming: A Case study for Typhoon Songda in 2016”. The current title is too broad, while this manuscript focuses on only one Typhoon case.

2.     Questions related to Methodology:

a.     In Eq (1), show the complete form of h(.)

b.     Based on your description in L149-154, -V_T^U is not used as all to determine the ET onset and completion. Please clarify how this term is used.

c.     L140: “All parameters are calculated within a 500-km radius from the cyclone center”: We know that when the SST increases, the TC size is greater (your SST4.5 experiment). 1) Do you think if it is more appropriate to use different radius to analyze your two experiments? 2) If using the same 500-km radius to analyze both experiments, do you expect any potential pitfalls to your analysis? Why?

d.     L179-180 “Among the forcing terms on …, RAF plays a dominant role in …”: you need evidence to back up this statement. Use either a reference or show one additional figure calculating contribution from each term in (5).

e.     What’s the intention of using two methods to determine ET phases? How is it related to the key message that you want the readers to get? My impression is that introducing two methods here only distract your readers from your main focus in this manuscript.

Minor comments:

L51: “in the simulations under future climate conditions”: What are general future climate conditions? Need to be clear here.

L52: “Jung and Lackmann (2021) use small ensemble to examine the response of North …”: How small? List the ensemble size here instead of saying “small ensemble”.

L58: “Their results show that under a more favorable background environment in the future”: What do you exactly mean as “more favorable background environment”? Be specific about it. Otherwise those summary of previous work are useless for the readers.

L83: unbold “m s^-1”

L156: “The other method is an alternate method”: “alternate” => “alternative”

L251-256: Try to avoid using (1), (2),… to list the differences unless you are introducing some steps in the algorithm. Here you can simply say “First, …. Besides, …”.

L431: “So the frontolysis”: Rephrase this sentence to avoid using “So” to start a sentence.

Reviewer 2 Report

The author raised the SST in the model region by 4.5 degrees, but the stratification of the whole atmosphere remains the same. This is not consistent with the actual atmospheric situation. Theoretically, the sea surface temperature changes and the atmosphere should change accordingly. Therefore, the size change in EXP SST4.5 cannot reflect the size change of TC during ET stage under projected warming. In addition, the author did not well diagnose the specific reasons for the change in the size of the typhoon during the ET process.

Major comments:

1. Directly increase SST by 4.5 degrees will promote convection near the eyewall due the reduced stratification. In fact, the atmosphere adjusts itself to the SST, so the whole stratification is changed accordingly. In this scenario, whether or not the strong convection near eyewall could generate in EXP4.5 still in question.   

2. The size change of TC is highly related to the convection in TC core region

3. It seems that at the early stages of ET, the frontal convection (north of TC) is more vigorous in Control simulation? Is there any explanation for this phenomenon?

4. I am not quite clear about the process of formation of cold pool north of TC center in SST4.5. Why strong cold pool is absence in Control simulation?

Minor comments:

1. The (a) and (b) are missing in Fig.4.

Reviewer 3 Report

In this study a sensitivity experiment is performed to examine the effect of raise in ocean temperatures to the characteristics of tropical cyclones that undergo Extratropical Transition. Typhoon Songda is used as an example, and two simulations are performed using WRF with different (real and artificially increased SSTs). The design of the experiment, the choice of the variables to be analysed, and the presentation of the results follows a remarkable order. The quality of the plots used, despite their complexity, makes it easy for the reader to follow the text. A lot of work has been invested in this study, that deserves to be published as is.

Reviewer 4 Report

Although much TC research has focused upon the TC intensity, less attention has been given to dynamics governing TC size. Previous studies have indicated that mean storm size is found to increase systematically with the relative sea surface temperature. The current paper seeks to examine another aspect of the TC size problem by examining how increased SST affects storm size during ET. The authors have shown that during ET, the baroclinicity-driven frontal convection in the outer region induces extensive boundary layer inflow, which expands the TC wind field. Furthermore, the authors posit a feedback mechanism in which increased SST leads to increased surface heat flux that strengthens the secondary circulation and the low-level inflow, leading to a larger TC size.

I believe that the authors have done significant work. Aside from some grammatical edits and other minor changes  (which I've uploaded to the PDF), I believe that this paper is suitable for publication.

Round 2

Reviewer 1 Report

The authors have addressed all my concerns with very detailed explanations. 

I recommend publication after 1 minor revision.

Thank you very much for sharing this interesting study.

 Minor comments:

L185: "when coarser grid spacing was the rule for analyses and simulations": change to "when TC simulations were more commonly conducted with a coarse-resolution "

Reviewer 2 Report

The authors has addressed all my concerns. Althought the  authors do not show the evolution of TC size in an adjusted tempreture profile, this may be explored in the authors' follow-up works.  

Author Response

Thank the reviewer very much for the positive evaluation and helpful comments.